# Role of Next Generation Immune Checkpoint Inhibitor (ICI) Therapy in Philadelphia Negative Classic Myeloproliferative Neoplasm (MPN): Review of the Literature

**DOI:** 10.3390/ijms241512502

**Published:** 2023-08-07

**Authors:** Ruchi Yadav, Narek Hakobyan, Jen-Chin Wang

**Affiliations:** 1Department of Internal Medicine, Brookdale University Hospital Medical Center, Brooklyn, NY 11212, USA; ryadav@bhmcny.org (R.Y.); nhakobya@bhmcny.org (N.H.); 2Department of Hematology/Oncology, Brookdale University Hospital Medical Center, Brooklyn, NY 11212, USA

**Keywords:** Myeloproliferative neoplasm, Immune checkpoint inhibitor, Philadelphia chromosome

## Abstract

The Philadelphia chromosome-negative (Ph−) myeloproliferative neoplasms (MPNs), which include essential thrombocythemia (ET), polycythemia vera (PV), and myelofibrosis (MF), are enduring and well-known conditions. These disorders are characterized by the abnormal growth of one or more hematopoietic cell lineages in the body’s stem cells, leading to the enlargement of organs and the manifestation of constitutional symptoms. Numerous studies have provided evidence indicating that the pathogenesis of these diseases involves the dysregulation of the immune system and the presence of chronic inflammation, both of which are significant factors. Lately, the treatment of cancer including hematological malignancy has progressed on the agents aiming for the immune system, cytokine environment, immunotherapy agents, and targeted immune therapy. Immune checkpoints are the molecules that regulate T cell function in the tumor microenvironment (TME). The first line of primary immune checkpoints are programmed cell death-1 (PD-1)/programmed cell death ligand-1 (PD-L1), and cytotoxic T-lymphocyte antigen-4 (CTLA-4). Immune checkpoint inhibitor therapy (ICIT) exerts its anti-tumor actions by blocking the inhibitory pathways in T cells and has reformed cancer treatment. Despite the impressive clinical success of ICIT, tumor internal resistance poses a challenge for oncologists leading to a low response rate in solid tumors and hematological malignancies. A Phase II trial on nivolumab for patients with post-essential thrombocythemia myelofibrosis, primary myelofibrosis, or post-polycythemia myelofibrosis was performed (Identifier: NCT02421354). This trial tested the efficacy of a PD-1 blockade agent, namely nivolumab, but was terminated prematurely due to adverse events and lack of efficacy. A multicenter, Phase II, single-arm open-label study was conducted including pembrolizumab in patients with primary thrombocythemia, post-essential thrombocythemia or post-polycythemia vera myelofibrosis that were ineligible for or were previously treated with ruxolitinib. This study showed that pembrolizumab treatment did not have many adverse events, but there were no pertinent clinical responses hence it was terminated after the first stage was completed. To avail the benefits from immunotherapy, the paradigm has shifted to new immune checkpoints in the TME such as lymphocyte activation gene-3 (LAG-3), T cell immunoglobulin and mucin domain 3 (TIM-3), T cell immunoglobulin and ITIM domain (TIGIT), V-domain immunoglobulin-containing suppressor of T cell activation (VISTA), and human endogenous retrovirus-H long terminal repeat-associating protein 2 (HHLA2) forming the basis of next-generation ICIT. The primary aim of this article is to underscore and elucidate the significance of next-generation ICIT in the context of MPN. Specifically, we aim to explore the potential of monoclonal antibodies as targeted immunotherapy and the development of vaccines targeting specific MPN epitopes, with the intent of augmenting tumor-related immune responses. It is anticipated that these therapeutic modalities rooted in immunotherapy will not only expand but also enhance the existing treatment regimens for patients afflicted with MPN. Preliminary studies from our laboratory showed over-expressed MDSC and over-expressed VISTA in MDSC, and in progenitor and immune cells directing the need for more clinical trials using next-generation ICI in the treatment of MPN.

## 1. Introduction

The myeloproliferative neoplasms (MPN), are characterized by the clonal proliferation of one or more hematopoietic cell lineages [1]. As per International Consensus Classification [2] (ICC) and World Health Organization (WHO), 5th edition [3] classic MPN comprises mainly essential thrombocythemia (ET), polycythemia vera (PV), and myelofibrosis (MF) [4]. Recently, there has been a notable shift in the treatment approach for MPN, with an increasing emphasis on agents that specifically target the immune system, cytokine milieu, immunomodulatory agents, and targeted immune therapy. The pathogenesis of MPN is not very clear but studies have shown that TNF-α selectively promotes the growth of JAK2V617F-positive MPN cells over controls contributing to clonal expansion of mutant copies during MPN progression [5]. Various driver mutations as studied by genetic sequencing, clonal analysis, and protein expression showed that tyrosine kinase Janus Kinase 2 -JAK2V617F mutation occurs in 95% of patients with PV and 50% to 60% of patients with ET and MF [6,7]. These mutations lead to hyperactive Janus kinase, signal transducer, and activator of transcription proteins (JAK-STAT) signaling pathways downstream of the erythropoietin receptor and thrombopoietin receptor (MPL) [8]. Concurrently profound immune dysregulation and defective immune surveillance also play a significant role in the pathogenesis of MPN [9]. The dysregulated genes associated with the immune system and inflammation, which have been implicated in myeloproliferative neoplasms (MPN), include the interferon-inducible gene [10], regulatory T cells (such as CD4+ CD25+ FOXP3+ Tregs) [11], natural killer cells, human leukocyte antigen (HLA) class I and II molecules, β2-microglobulin, molecules involved in the processing of HLA I antigens (such as LMP2, LMP7, TAP1/2, and tapasin) [10,12], as well as antioxidative stress genes (ATM, TP53, CYBA, NRF2, PTGS1, and SIRT2) [13,14]. Furthermore, the upregulation of immunosuppressive cells, such as myeloid-derived suppressor cells (MDSCs), contributes to the evasion of tumor cells from immune surveillance, thus playing a crucial role in the etiopathogenesis of MPN [15]. The key events involved in the development of the neoplastic process are oncogenic transformation and immune escape allowing for uncontrolled proliferation and avoidance of apoptosis. Immune checkpoint inhibitory therapy (ICIT) is based on blocking the inhibitory pathways in T cells to promote anti-tumor immune responses and has revolutionized cancer treatment paradigms. Oncogenic JAK2 activation results in high expression of programmed death-ligand 1 (PD-L1) on the surface of monocytes, megakaryocytes, MDSC, and platelets which is mediated via the JAK2-STAT3 and JAK2-STAT5 axes [16]. The blocking of the PD-1 pathway, which was found to be overexpressed in myeloid malignancies, has gained immense interest as a therapeutic target paving the path for novel strategies. One such trial was ClinicalTrials.gov Identifier: NCT02421354 where the efficacy and safety of single-agent nivolumab (PD-1 inhibitor) in eight adult patients with myelofibrosis was tested [17]. However, this study was terminated early due to failure to meet the predetermined efficacy endpoint. At the American Society of Hematology (ASH) annual meeting in 2020, a multi-center, open-label, Phase II, single-arm study of pembrolizumab was presented with its use in patients with primary, post-essential thrombocythemia or post-polycythemia vera MF (NCT03065400) [18]. Nine cases were presented, but none had a clinical response.

The use of ICI in hematological malignancies brings a daunting challenge with a low response rate thus letting the oncologist/molecular physicians change the focus to dig deeply into the tumor microenvironment for alternative therapeutic targets. To permit more patients to benefit from immunotherapy, the focus has changed to targeting alternative novel immune checkpoints in the tumor microenvironment such as lymphocyte activation gene-3 (LAG-3), T cell immunoglobulin, mucin domain 3 (TIM-3), T cell immunoglobulin, ITIM domain (TIGIT), V-domain immunoglobulin-containing suppressor of T cell activation (VISTA) and human endogenous retrovirus-H long terminal repeat-associating protein 2 (HHLA2) forming the basis of next-generation ICIT [19] as shown in Figure 1. The primary objective of our review article is to highlight and explore the role of next-generation ICI in the context of MPN. Specifically, we aim to emphasize the potential of targeted immunotherapy utilizing monoclonal antibodies, checkpoint inhibitors, and therapeutic vaccines directed against specific MPN epitopes. The intended outcome of these approaches is to further enhance tumor-specific immune responses in MPN patients.

## 2. LAG-3 Targeted Therapy and Its Role in Hematological Malignancies

Lymphocyte activation gene 3 (LAG-3) (CD223) is a CD4-related activation-induced cell surface inhibitory receptor that binds to major histocompatibility complex (MHC) class II molecules with high affinity and negatively regulates T cell effector functions [20]. Cells expressing LAG-3 are T cells, some activated B cells, plasmacytoid dendritic cells (DCs), and neurons [21]. LAG-3 ligands are MHC class-II, galectin-3 (Gal-3), and fibrinogen-like protein 1 (FGL1) with MHC-II being the canonical ligand [22]. LAG-3 binds to MHC class II with higher affinity than CD4 inducing rapid protein phosphorylation of phospholipase Cgamma2 (PLCgamma2) and p72syk as well as activation of phosphatidyl inositol 3-kinase/Akt, p42/44 extracellular signal-regulated protein kinase, and p38 mitogen-activated protein kinase pathways [23]. Gal-3 is expressed on tumor cells and activated T cells that are needed for CD8 T cell and plasmacytoid DC suppression [22]. FGL1 is highly produced by human cancer cells and binding of LAG-3 with FGL1 contributes to poor responses/resistance to anti-PD-1/anti-PD-L1 immunotherapies [24,25]. This mechanism forms the basis of PD-1 and LAG-3 co-blockade responsible for several T cell antitumor activities [26,27,28].

Currently, 16 LAG-3 targeted therapies are tested at 97 clinical trials by Bristol-Myers Squibb (BMS-986016), Regeneron Pharmaceuticals (REGN3767 and 89Zr-DFO-REGN3767), Merck (MK-4280), Novartis (LAG525), Tesaro (GSK) (TSR-033), Symphogen (Sym022), GlaxoSmith (GSK2831781), Incyte Biosciences International Sàrl (INCAGN02385), Prima BioMed/Immutep (IMP321), MacroGenics (MGD013), F-Star (FS118), Hoffmann-La Roche (RO7247669), Shanghai EpimAb Biotherapeutics (EMB-02), Xencor (XmAb841) and Innovent Biologics (IBI323) [29]. LAG-3 targeted therapies are divided into three categories namely monoclonal antibodies, soluble LAG-3—immunoglobulin (Ig) fusion proteins, and anti-LAG-3 bispecific drugs [29]. Most trials are Phase II (34), I/II (21), and II (35), and only two of them have reached Phase III for BMS-986016 (NCT05002569) [30] and MK-4280 drugs (NCT05064059) [31]. Table 1 demonstrates the use of LAG-3 agents in hematological malignancies in the current clinical trials.

Use of 89Zr-DFO-REGN3767 in PET scans of people with diffuse large B cell lymphomas (DLBCL) was the pilot study (NCT04566978) [32] undertaken at Memorial Sloan Kettering Hospital in 2022 with the main purpose of the study looking at the way the body absorbs, distributes, and gets rid of 89Zr-DFO-REGN3767 [33]. 89Zr-DFO-REGN3767 is comprised of the anti-LAG-3 antibody, REGN3767 labeled with the positron-emitter zirconium-89 (89Zr) through the chelator-linker DFO and REGN3767 is an investigational monoclonal antibody that targets LAG-3 receptors. This study is a diagnostic research study determining the optimal time for imaging and tumor uptake post 89Zr-DFO-REGN3767 administration. However, it can help evaluate tumor lesion uptake of 89Zr-DFO-REGN3767 and correlate with LAG-3 expression by immunohistochemistry (IHC) in tumors that will be compared descriptively with other biomarkers of TME characterized in biopsies, such as quantitation of IHC score (LAG-3 and/or other immune cell markers).A safety and efficacy trial of JCAR017 (lisocabtagene maraleucel, also known as liso-cel) (a CD19-targeted chimeric antigen receptor CART-cell therapy) combinations in subjects with relapsed/refractory B cell malignancies (PLATFORM) (NCT03310619) [34] was performed. Relatlimab, BMS-986016 is an anti-LAG-3 fully human monoclonal IgG4-κ antibody that binds human LAG-3 with high affinity and inhibits its binding to MHC-II [35]. This trial was a global, open-label, multi-arm, parallel multi-cohort, multi-center, Phase I/II study to determine the safety, tolerability, pharmacokinetics, efficacy, and patient-reported quality of life of JCAR017 in combination with various agents including relatlimab, durvalumab, avadomide, iberdomide, ibrutinib, and nivolumab. The trial was completed, and the studied tumors were non-Hodgkin lymphoma (NHL), diffuse large B cell lymphoma (DLBCL), and Follicular lymphoma (FL). The objective of this study was that during Phase I, different arms may be opened to test JCAR017 in combination with combination agent(s) in adult subjects with R/R aggressive B cell NHL. Within each arm, different doses and schedules of JCAR017 and the combination agent(s) may be tested in several cohorts and subcohorts per arm. During Phase II of this study, the expansion of any dose level and schedule for any arm that is safe may occur. All subjects from Phase I and Phase II will be followed for 24 months for survival, relapse, long-term toxicity, and viral vector safety as per guidelines.A similar trial was also designed with relatlimab by Bristol-Myers Squibb, NCT02061761 [36] administered alone or in combination with nivolumab to subjects with relapsed or refractory B cell malignancies (relapsed or refractory Hodgkin lymphoma (HL) and relapsed or refractory DLBCL and to study its safety, tolerability, dose-limiting toxicities and maximum tolerated dose. The trial completed and studied hematological malignancies including chronic lymphocytic leukemia (CLL), HL, NHL, and Multiple Myeloma (MM). A detailed description of dose-related adverse events was studied and was +displayed in the result section of the trial.Favezelimab (MK-4280) is another LAG-3 antibody that is studied in combination with pembrolizumab (MK-3475) in the clinical trial NCT03598608 [37] that was started in July 2018 to study and evaluate the safety and efficacy of these agents in hematologic malignancies. It included classical HL, DLBCL, and indolent HL. No results have been posted till the writing of this article. This study will also evaluate the safety and efficacy of pembrolizumab or favezelimab administered as monotherapy in participants with classical HL using a 1:1 randomized study design.Relapsed or refractory acute myeloid leukemia (AML) and newly diagnosed older AML are included in the ClinicalTrials.gov Identifier: NCT04913922 [38] to study the combination of relatlimab with nivolumab and 5-azacytidine. No results have been posted yet.

All the above trials included LAG-3 as an ICI agent in the above-mentioned hematological malignancies, however, no trials have been conducted in the field of MPN. We unfold the mechanism of action of LAG-3 to provide a better understanding of its potential use in the future as depicted in Figure 2.

## 3. Mechanism of Action of LAG-3

In 1990, Triebel and colleagues discovered LAG-3, a novel type I transmembrane protein consisting of 498 amino acids, which is expressed on activated human natural killer (NK) and T cell lines [39]. The LAG-3 gene is located in close proximity to CD4 on chromosome 12 in humans (chromosome 6 in mice). Structurally, LAG-3 exhibits high similarity to CD4, featuring four extracellular immunoglobulin superfamily (IgSF)-like domains (D1-D4) [40]. These structural motifs are conserved between LAG-3 and CD4, resulting in similar extracellular folding patterns. Consequently, LAG-3 can bind to major histocompatibility complex (MHC) class II molecules, albeit at a distinct site, with even greater affinity than CD4 [41]. LAG-3 was suggested to be spatially associated with the T cell receptor TCR: CD3 complex present in lipid raft microdomains to allow for the clustering of signaling molecules and the formation of the immunological synapse. However, the exact mechanism is still unclear [42]. LAG-3 lacks a binding site in the cytoplasmic tail for the tyrosine kinase p56^Lck^, which CD4 uses to promote signal transduction downstream of the T cell receptor (TCR) [41]. Instead, the LAG-3 cytoplasmic domain appears to have three well-defined motifs namely a serine-based motif that could act as a PKC substrate, a repetitive “EP” motif consisting of a series of glutamic acid-proline dipeptide repeats, and a relatively unique “KIEELE” motif, highlighted by an essential lysine residue [43,44]. The absence of the cytoplasmic tail in LAG3 mutants reveals an intriguing aspect of its function, as these mutants neither compete with CD4 nor mediate the inhibitory effects typically associated with LAG3 [20]. This observation highlights the importance of transmitting an inhibitory signal through LAG3’s cytoplasmic domain. Notably, the expression of MHC class II molecules on human melanoma cells is linked to unfavorable prognoses. In the context of melanoma-infiltrating T cells, the high expression of LAG3 and its interaction with MHC class II molecules may contribute to clonal exhaustion [45]. This interaction, demonstrated in vitro, could potentially serve as an evasion mechanism employed by tumor cells, safeguarding them against apoptosis. Recent studies indicate that melanoma cells expressing MHC class II molecules attract a specific infiltration of CD4+ T cells, potentially facilitated by the interaction between LAG3 and MHC class II molecules. Consequently, this interaction negatively impacts CD8+ T cell responses [46,47]. These findings shed light on the intricate interplay between LAG3, MHC class II molecules, and tumor cells, ultimately influencing immune responses and highlighting LAG3 as a key modulator in cancer immunology. Galectin-3, a ligand expressed by numerous cells within the tumor microenvironment rather than the tumor itself, has the potential to interact with LAG3 on tumor-specific CD8+ T cells, thereby modulating anti-tumor immune responses [48]. Another interesting molecule, liver sinusoidal endothelial cell lectin (LSECtin), is present in the liver and has also been detected in human melanoma tissues, where it facilitates tumor growth by inhibiting T cell-dependent anti-tumor responses [49]. Interaction between LAG-3 and LSECtin in melanoma cells has been found to impede IFNγ production by antigen-specific effector T cells, thereby altering the tumor microenvironment [49]. The persistence of T cell activation within a chronic inflammatory environment, particularly in the presence of tumors, often leads to the sustained co-expression of LAG3 and other inhibitory receptors (IR) such as PD1, TIGIT, TIM3, CD160, and 2B4, resulting in a state of T cell dysfunction [50]. Although LAG3 is widely expressed across various hematopoietic cell types, including CD11clow B220+ PDCA-1+ plasmacytoid dendritic cells (pDCs) which exhibit higher levels of LAG3 expression compared to other subsets, it is not expressed on any myeloid or lymphoid DC subset [51]. In vitro experiments have demonstrated that MHC class II-expressing melanoma cells can stimulate LAG3+ pDCs to mature and produce IL-6, a finding confirmed in vivo where LAG3+ pDCs displayed elevated IL-6 production and an activated phenotype in close proximity to melanoma cells [52]. Furthermore, a study by Bo Huang et al. suggests that increased IL-6 levels prompt the release of CCL2 by monocytes in vitro, which in turn may recruit MDSCs, thus proposing the hypothesis that LAG3+ pDCs may indirectly drive MDSC-mediated immunosuppression through engagement with MHC class II+ melanoma cells [53]. The activity of LAG-3 is also regulated through cell surface cleavage mediated by ADAM10 and ADAM17 disintegrin/metalloproteases. However, it is important to note that soluble LAG-3 does not seem to possess any biological function in mice [54].

## 4. V-Domain Immunoglobulin Suppressor of T Cell Activation (VISTA) Targeted Therapy and Its Role in Hematological Malignancies

VISTA (also known as c10orf54, VSIR, SISP1, B7-H5, PD-1H, DD1α, Gi24, and Dies1) is primarily expressed in myeloid cells, particularly microglia, and neutrophils followed by monocytes, macrophages, and dendritic cells [55,56]. Moreover, VISTA demonstrates its highest expression levels on naïve CD4+ T cells and Foxp3+ regulatory T cells [57]. Structurally, VISTA is a type I transmembrane protein characterized by a single N-terminal immunoglobulin (Ig) V-domain, sharing its greatest homology with PD-L1 [58]. The precise role of VISTA in immune regulation remains intricate and not entirely elucidated. Notably, VISTA acts not only as a ligand expressed on antigen-presenting cells but also functions as a receptor on T cells [59]. Most studies conducted thus far have focused on describing the inhibitory impact of VISTA on the immune system and have shown that VISTA deficiency or anti-VISTA treatment can effectively enhance immune responses [60]. Owing to its predominant expression on macrophages, VISTA has been implicated as a potential target for immunotherapy in melanoma [61]. Studies have indicated a correlation between melanoma survival and the expressions of PD-L1 and VISTA [62,63]. Furthermore, tumor cell-specific expression of VISTA, regulated by the forkhead box D3 (FOXD3) factor, promotes tumorigenesis and enhances PD-L1 expression on tumor-infiltrating macrophages in vivo, which is associated with increased intra-tumoral T regulatory cells [62]. It is worth noting that VISTA exhibits high expression on myeloid-derived suppressor cells in the peripheral blood, and there is a strong positive correlation between MDSC expression of VISTA and T cell expression of PD-1 in patients with AML, although direct regulation has yet to be substantiated [64,65]. MDSCs are myeloid cells that are defined into subsets namely monocytic MDSCs (CD15^−^) and granulocytic MDSCs (CD15^+^) [66]. Patients with AML displayed increased expression of VISTA on MDSCs highlighting the role of VISTA in MDSC-mediated CD8 T cell response [64]. Conflicting evidence exists regarding the role of VISTA, with certain studies suggesting that it functions as an immune checkpoint receptor expressed on tumor-infiltrating T lymphocytes (TILs) and myeloid cells, leading to the suppression of T cell activation, proliferation, and cytokine production [67,68]. Conversely, other studies have demonstrated that VISTA is overexpressed in tumor tissues and operates as a co-stimulatory molecule [69,70].

Currently, clinical trials of VISTA-targeted cancer therapy are in progress namely ClinicalTrials.gov Identifier: NCT02671955 [71] and ClinicalTrials.gov Identifier: NCT02812875 [72]. JNJ-61610588 (CI-8993) [71] is a human monoclonal antibody against VISTA with potential negative checkpoint regulatory and antineoplastic activities that is being studied in advanced cancer patients. No study results have yet been posted. Presently, there is an ongoing study investigating CA-170 [72], a small drug-like molecule inhibitor specifically targeting PD-L1 and VISTA, in patients with advanced solid tumors or lymphomas. However, it is important to note that the trial coordinators are not currently recruiting participants, and the most recent update regarding the trial was posted on 6 May 2019. There are pre-clinical trials of VISTA mentioned in hematological cancer and solid tumors involving IGN-381 (mAbs by Ingenica Biotherapeutics) and HMBD-002 (mABs by Hummingbird Bioscience) [73]. HMBD-002 exerted significant inhibitory effects on tumor progression and its combination with anti-PD-L1 was found to be more effective in tumors that showed abundant MDSC infiltration [74].

The Tumor and Immune System Interaction Database (TISIDB) [69,75] undertook a comprehensive analysis to examine the potential relevance of VISTA in the context of cancer immunity across an expansive array of 30 distinct cancer types. The findings unveiled the following insightful outcomes: Firstly, there existed a positive correlation between the expression levels of VISTA and nearly all categories of TILs that possessed either tumor-suppressing or tumor-promoting functions. Noteworthy TIL subpopulations encompassed, among others, activated CD8 T cells, natural killer (NK) cells, Tregs, and MDSCs. Secondly, an affirmative association emerged between the expression levels of VISTA and the relative abundance of a vast majority of critical immunomodulators. Notably, this correlation extended across the diverse functionalities of immune inhibitors, immunostimulators, and MHCs within the context of the examined 30 cancer types. Prominent immunomodulators in this regard included pivotal immune checkpoint components such as PD-1, PD-L1, CD80, and CD86. Lastly, VISTA expression demonstrated a positive correlation with the relative abundance of widely recognized chemokines and their corresponding receptors across the expansive spectrum of 30 cancer types. Notable chemokines in this context comprised CXCL1, CXCL8, CXCL10, and CXCR3, among others. These findings collectively highlight the potential of VISTA to function both as a receptor and a ligand, thereby engaging distinct partners to modulate immune responses. It is evident that VISTA modulation holds promise as a compelling therapeutic target, warranting further investigation, particularly in hematological cancers, including MPN. Moreover, by targeting VISTA to alleviate suppression exerted by myeloid cells, the efficacy of T cell-focused therapies, such as anti-PD1 and anti-CTLA4, could potentially be enhanced, particularly in cases of monotherapy resistance observed with other ICIT.

## 5. Role of T Cell Immunoglobulin and Mucin Domain 3 (TIM-3) as Next-Generation ICI in Hematological Malignancies

T cell immunoglobulin and mucin domain 3 (TIM-3) is a type I transmembrane protein that was first discovered on terminally differentiated CD4+ type I helper T cells (TH1 cells) and CD8+ cytotoxic T cells (CTLs) [76]. Later on, its expression was observed on other T cell subtypes, excluding TH2 cells, as well as some other immune cells including dendritic cells (DCs), NK cells, macrophages, monocytes, mast cells, and some malignant cells [77,78,79,80]. Tim-3 is constitutively expressed on DCs and macrophages in both humans and mice, specifically in humans where it suppresses IL-12 expression [81,82]. In DCs, Tim-3 inhibits its activation and maturation by blocking NF-κB signaling via a Btk-c-Src signaling-dependent mechanism, interfering with the ability of cytoplasmic toll-like receptors (TLRs) to sense immunogenicity and thereby suppressing anti-tumor immunity [83]. There are four known ligands for TIM-3 namely galectin-9 (Gal-9) [84]—which has been reported to induce apoptosis in TH1 cells, High-mobility group protein B1 (HMGB1) [85]—known as “alarmin”, which is released from damaged cells and induces the activation of phagocytes, phosphatidylserine (PtdSer) [86]—“eat-me” signal induction molecule and carcinoembryonic antigen cell adhesion molecule 1 (CEACAM-1) [87]—which can have both *cis* and *trans* interactions with TIM-3. Interactions between Tim-3 with its ligands, galectin-9 and Ceacam-1, resulting in phosphorylation of tyrosine residues namely Y_256_ and Y_263_, and release of HLA-B associated transcript 3 (Bat3) from the Tim-3 tail, thereby promoting Tim-3-mediated T cell inhibitory function by allowing binding of SH2 domain-containing Src kinases and subsequent regulation of TCR signaling [88,89]. Studies have reported that a higher percentage of TIM-3 reflects a higher risk for myelodysplastic syndrome (MDS) transformation to leukemia as increased levels of TIM-3 and its ligand, Gal-9 is reported on bone marrow cells and MDSCs from MDS patients [90,91]. This highlights the TIM-3/Gal-9 axis role in the proliferation of blasts, induction of immune escape, and T cell exhaustion supporting disease progression [92]. Bruck et al. observed TIM-3 overexpression on exhausted CD4+ and CD8+ T cells in untreated chronic myeloid leukemia (CML) patients and reported a correlation between PD-1+ TIM-3- CD8+ T cells and poor response to Tyrosine kinase inhibitors (TKIs) [93]. Dysfunctional immunity plays a major role in malignancy formation, but many more clinical studies are required to examine the expression of TIM-3 in MPN and other immune cell types and establish its role in the formation, therapy resistance, relapse, and immune scoring in this malignancy. Several clinical trials involving TIM-3 blockade, especially in combination with PD-1 blockade, have demonstrated reliable preliminary results against solid tumors namely HBV-related hepatocellular carcinoma [94,95,96]. TIM-3 is highly expressed in peripheral blood and bone marrow exhausted T cells in various hematological malignancies, including acute lymphoblastic leukemia (ALL), chronic lymphocytic leukemia (CLL), and multiple myeloma (MM) however, few reports have demonstrated its clinical significance as monotherapy with TIM-3 inhibitors alone [97,98,99].

Further studies are required to evaluate the efficacy of TIM-3 inhibitors in different types and stages of leukemias and MPNs concerning bone marrow microenvironments. Currently, the TIM-3 inhibitors used in clinical trials include MBG453 (also known as sabatolimab), TSR-022 (Tesaro), BMS-986258, LY3321367, SYM023, BGB-A425, and SHR-1702 [100,101]. Currently, sabatolimab (high-affinity IgG4 mAb) is the only anti–TIM-3 mAb being investigated in MDS and acute myeloid leukemia (AML) with preliminary safety and efficacy data. ClinicalTrials.gov Identifier: NCT03066648 is an active Phase I trial of TIM-3 involving the study of PDR001 and/or MBG453 in combination with decitabine or azacitidine in patients with AML or high-risk MDS [102]. It includes AML, MDS, chronic myelomonocytic leukemia, and bone marrow diseases. No result was posted until the writing of this article, but preliminary results reported that the combination of sabatolimab plus HMA (either decitabine or azacitidine) was associated with mostly grade 1 or 2 drug-related AEs and showed preliminary evidence of antileukemic activity and response durability. Based on preliminary follow-up, overall response rates (ORRs) in patients with HR-MDS with sabatolimab plus decitabine and sabatolimab plus azacitidine were 61.1% and 57.1%, respectively, with complete response (CR) rates of 33.3% and 7.1% [100]. TIM-3 is relatively higher expressed on leukemic stem cells than non-neoplastic hematopoietic stem cells, often in conjunction with other surface antigens such as CD33, CD123, and CLL, thus targeting TIM3 might be a novel approach in cancer treatment in the future [103]. Targeting Tim-3 along with other checkpoint inhibitors or combining Tim-3 inhibition with new immunotherapeutic approaches that activate cancer-specific T cell stimulatory molecules have immense potential for developing modalities with durable clinical benefits.

## 6. Role of T Cell Immunoglobulin and Immunoreceptor Tyrosine-Based Inhibitory Domain (TIGIT) as a Target for Next-Generation ICI in Hematological Malignancies

TIGIT belongs to a family of PVR-like proteins first discovered in 2009. It is composed of one extracellular immunoglobulin variable domain, a type I transmembrane domain, a short intracellular domain containing an immunoreceptor tyrosine-based inhibitory motif (ITIM), and an immunoglobulin tyrosine tail (ITT)-like motif [104,105]. TIGIT, also known as Washington University cell adhesion molecule (WUCAM), is expressed on T cells and natural killer (NK) cells, and it shares CD155 (also known as poliovirus receptor, nectin, and nectin-like 5 [NECL-5]) as a ligand along with DNAX accessory molecule-1 (DNAM-1) and CD96 [106,107]. The immunoglobulin variable domain of TIGIT exhibits sequence homology with other members of the PVR-like family, including DNAM-1, CD96, CD111, CD155, CD112 (also known as PVR-related 2 [PVRL2], nectin-2), and CD113 (also known as poliovirus receptor-related 3 [PVRL3], nectin-3), as well as PVRL4 [104]. In both humans and mice, the primary ligand for TIGIT is CD155, while it binds with relatively lower affinity to CD112 and CD113 [105,108]. CD155 is mainly expressed in DCs, T cells, B cells, macrophages, kidneys, the nervous system, and intestines [109]. CD112 has a wide expression in bone marrow, kidneys, pancreas, and lungs [110]. CD113 is restricted to non-hematopoietic tissues, including the placenta, testis, kidneys, liver, and lungs [111]. The mechanism of action of TIGIT may have a cell-extrinsic pathway, as a ligand for CD155 [104] or a cell-intrinsic pathway by interfering with DNAM-1 co-stimulation [112,113] or by directly delivering inhibitory signals to the effector cell [105]. Following ligation of TIGIT, the signaling of CD155 in human monocyte-derived dendritic cells (DCs) induces an elevation in IL-10 secretion while concurrently reducing the secretion of the proinflammatory cytokine IL-12. This effect contributes to the promotion of tolerogenic DCs, which subsequently downregulate T cell responses [106]. Regarding the cell-intrinsic mechanism of action, it has been hypothesized that due to the high affinity of TIGIT for CD155, TIGIT may exert its inhibitory effects on T cells by competitively binding to CD155, thereby outcompeting DNAM-1. This proposition was initially suggested based on observations that knockdown of TIGIT in human CD4+ T cells resulted in increased expression of T-bet and IFN-γ. Furthermore, this effect could be counteracted by blocking DNAM-1 [112,113]. Notably, upregulation of TIGIT has been observed in various malignancies, including melanoma, breast cancer, non-small-cell lung carcinoma (NSCLC), colon adenocarcinoma, gastric cancer, multiple myeloma (MM), and acute myeloid leukemia (AML) [114,115,116,117,118].

In mouse pre-clinical models and cancer patients, TIGIT expression on tumor-infiltrating CD8^+^ T cells often correlates with increased expression of other inhibitory receptors such as PD-1, LAG-3, TIM-3, and with decreased expression of DNAM-1 [115,119,120,121]. Similarly, a high TIGIT/DNAM-1 ratio on tumor-infiltrating T_regs_ was shown to correlate with poor clinical outcomes following ICB targeting PD-1 and/or CTLA-4 [122]. In the pre-clinical mice TIGIT negative mice bearing colon cancer (MC38 model), co-blockade of TIGIT and PD-1 was associated with enhanced effector cell functions of both CD4^+^ and CD8^+^ T cells compared to either therapy alone; and TIGIT/PD-1 co-blockade produced a 100% cure rate [123].

As previously discussed, tumor cells can establish an immune-suppressive microenvironment through various mechanisms, including the secretion or promotion of immunosuppressive cytokines such as interleukin (IL)-10, transforming growth factors (TGF)-β, the recruitment of regulatory cells such as regulatory T cells (Tregs), myeloid-derived suppressor cells (MDSCs), and type II macrophages, as well as the modulation of immune cell metabolism [124,125]. However, immune checkpoint pathways, which consist of receptor-ligand pairs, play a crucial role in suppressing the effector functions of T cells and natural killer (NK) cells, thereby impairing anti-tumor immunity [126]. Despite the remarkable success of immune checkpoint inhibitor therapies (ICIT), a considerable number of patients fail to respond to current immunotherapies or experience treatment-related toxicities known as “immune-related adverse events” (irAEs), which can sometimes be fatal [127,128]. Consequently, there is a growing interest in identifying novel immune checkpoints that can be targeted effectively with high anti-tumor efficacy while potentially minimizing irAEs across various malignancies. A T cell immunoreceptor with Ig and ITIM domains (TIGIT) has emerged as a compelling negative regulator of cytotoxic lymphocytes and represents an attractive target for cancer therapy, potentially offering a reduced risk of irAEs compared to anti-PD-1 or anti-CTLA-4 monoclonal antibodies (mAbs) [129,130].

Presently, six human clinical trials of anti-TIGIT-mAb of IgG1 isotype are undergoing including etigilimab (OMP-313M32), in Phase I/II, either as monotherapy or combinations with PD-1/PD-L1 blockade, for the treatment of solid cancers [131,132,133,134,135,136].

TIGIT is highly expressed on tumor-infiltrating lymphocytes (TIL) in several hematological malignancies including follicular lymphoma, CLL, classic HL, AML, and relapsed MM, helping in tumor progression and poor outcome [137]. Research studies have shown the immense potential of anti-TIGIT therapy as reported by Catakovic’s in vitro experiment showing reduce CLL viability by TIGIT blockade [138]. Anti-TIGIT treatment prevented T cell exhaustion and prolonged survival in MM mice [139]. Current clinical trials based on therapeutic strategies targeting TIGIT have encouraging efficacy in hematological malignancies [140,141,142,143,144,145,146,147,148]. Table 2 shows current clinical trials of anti-TIGIT antibodies in hematological malignancies.

## 7. The Current and Future Role of ICI in the Management of MPN

The management of MPNs is constantly evolving and highly individualized. The optimal management of patients with MPNs necessitates intricate decision-making processes that take into account various factors, including the specific type of the disease, individual prognosis, age, and comorbidities, as well as the risks and benefits associated with available therapeutic options. Patients with PV and ET include thrombotic risk reduction using aspirin as well as cytoreduction with hydroxyurea (HU) or interferon-based therapy [149,150]. Cytoreductive therapy is selectively employed for individuals with high-risk diseases, such as those who meet criteria such as age over 60 years, a history of previous thrombosis, or a JAK2 mutation as assessed by the revised IPSET score [151]. HU is associated with significant side effects and subsequently, 24% of patients with PV or ET develop resistance to primary therapy necessitating the need for second-line therapy [152]. Interferon is frequently used as a frontline or second-line therapy including a novel, mono-pegylated formulation called Ropeginterferon alfa-2b, the first and only approved treatment for PV independent of previous hydroxyurea exposure [150,153]. The progress made in molecular science, particularly the identification of the Janus kinase 2 (JAK2) V617F mutation and its involvement in the dysregulation of the JAK-STAT pathway, has led to the emergence of the JAK inhibitor ruxolitinib. This therapeutic development has become the foundation of medical treatment for MF and PV cases that are resistant or intolerant to hydroxyurea [150]. Furthermore, the JAK1/2 inhibitor ruxolitinib is approved in intermediate to high-risk MF, as well as advanced PV after HU intolerance or failure [154]. JAK inhibitors can reduce symptom burden, and improve performance status and disease-associated cachexia is responsible for adding the survival benefit of these drugs [155]. Long-term follow-up studies showed improvement in bone marrow morphology (approximately 50% of patients may experience partial regression in marrow fibrosis over a period of 60 months) [156]. However, complete molecular remissions are infrequent, with only three patients in the RESPONSE-I trial and six patients in the COMFORT-I trial achieving such remissions [157,158]. The major limitations of the use of these agents are that they have debatable disease-modifying activities, there is the likelihood of losing response over time (median duration of spleen response 3 years), development of treatment resistance, on-target anemia, and thrombocytopenia stemming from JAK2 inhibition frequently limiting optimal dosing [159].

Ongoing research efforts are dedicated to improving the efficacy and safety of established treatment modalities as well as characterizing novel therapeutic approaches, many of which target the immune system. Lately, the focus of the treatment of MPN is based on the agents targeting the immune system, cytokine milieu, immunomodulatory agents, and targeted immune therapy. At the American Society of Hematology (ASH) annual meeting in 2020, Mascarenhas et al. [18] presented a multi-center, open-label, Phase II, single-arm study of pembrolizumab in patients with primary, post-essential thrombocythemia or post-polycythemia vera myelofibrosis (MF) (NCT03065400). Nine case studies were presented, but none had a clinical response. Wang et al. published an article demonstrating that PD-1 and PD-L1 were increased in MPN disease in immune cells, including CD4, CD8, monocyte, and CD34+ cells [160]. The potential stimulators of PD-L1 expression are interferon-gamma (IFN-ϒ), IL-10, VEGF, and hypoxia leading to activation of PD-L1 transcription [161,162]. Treg cells can stimulate B7-H1 expression in MDSCs thus enhancing each other’s immune suppression functions [163]. The role of MDSCs in the tumor microenvironment is getting defined day by day and they are implicated in inducing therapeutic resistance to ICI therapy [164,165]. Further studies summarized that abnormal MDSC accumulation in patients with advanced melanoma, non-small cell lung cancer, and breast cancer led to resistance to immunotherapy with a strong positive correlation between the MDSC percentage and neutrophil/lymphocyte rate (NLR) (a prognostic marker in both ipilimumab and nivolumab therapy) [166,167,168,169]. ICI targeting PD-1 stimulated circulating Treg levels but did not change Granulocyte-MDSC (G-MDSC) and Myeloid-MDSC (M-MDSC) levels. However, the partial response (PR) group had a higher baseline level of M-MDSCs, which exhibited a significant decrease after the first cycle of anti-PD-1 treatment [170]. Therefore, MDSC accumulation plays a significant role within the tumor microenvironment and is implicated in the failure of ICI.

There have been limited studies on the use of ICI in the treatment of MPN as described earlier in the article with three NCI-sponsored clinical trials related to combined immune- therapy (NCT03065400, NCT02421354, and NCT02871323) in 2021 [18,19,171].

We have collected preliminary data in our laboratory showing the expressions of VISTA, TIM-3, and LAG-3 on the progenitor, immune, and MDSC cells in MPN patients. We found that VISTA is the predominant next ICI receptor or ligand found in MPN patients. Other next-generation checkpoints including TIM-3, TIGIT, and LAG-3 were not different in expressions between controls and MPN patients as shown in Figure 3, Figure 4, Figure 5 and Figure 6. We had previously found MDSC over-expressed in cells including CD34^+^, CD14^+^, CD4^+^, and CD8^+^, and now our preliminary data suggest that VISTA (one of the next generation ICI) as compared to others such as TIM-3, LAG-3, TIGIT could be the predominant ICI target in MPN.

## 8. Future Directions and Perspectives

The integration of immunotherapy into the standard treatment regimens for a diverse array of tumor types, particularly hematological malignancies, has garnered considerable interest owing to significant advancements in targeting immune checkpoints through next-generation ICIT.

Firstly, the optimal combination of immunotherapeutic modalities remains a subject of substantial debate despite the potential for substantial therapeutic enhancements through combination immunotherapy. Secondly, while immune checkpoint blockade can yield remarkable tumor regression and remission, this response is observed in only a subset of patients. It is imperative to comprehend the underlying reasons for this limited efficacy. Although factors such as immune checkpoint ligand expression, temporal aspects of neoantigen expression, and their immunogenicity may contribute to this phenomenon, there likely exist numerous unknown factors. Unraveling these elusive factors and identifying biomarkers capable of predicting responsiveness to specific immunotherapeutic modalities will assume paramount importance. Thirdly, while the ongoing clinical trials investigating current immunotherapies constitute a significant stride forward, it remains imperative to sustain the swift development of new therapeutic targets in the research pipeline. These novel targets may serve as vital components of future combinatorial approaches, particularly if they exhibit similar efficacy but with reduced adverse events. Furthermore, continuous efforts to discover novel potential immunotherapeutic targets, mechanisms, and innovative delivery platforms are of utmost importance.

Our preliminary results showed that VISTA and others including TIM-3, LAG-3, and TIGIT, were the predominant next-generation ICI expressed on CD3^+^, CD14^+^, and CD34^+^ cells as measured by the percentage of positive 2nd G- ICI cells (Figure 3) and MFI (Figure 4). Furthermore, we demonstrated that VISTA also wares the predominant 2nd G-ICI on both the G-MDSC and M-MDSC as measured by the percentage of positive cells (Figure 5) and MFI (Figure 6) respectively. This would lead to further clinical trials specifically involving VISTA with a possible combination anti-Vista and anti-PD-1 in MPN disease. This may lead to reviving the ICI therapy in MPN which ICI was found to be a negative trial in using anti-PD-1 only in the treatment of MPN.

## Figures and Tables

**Figure 1 ijms-24-12502-f001:**
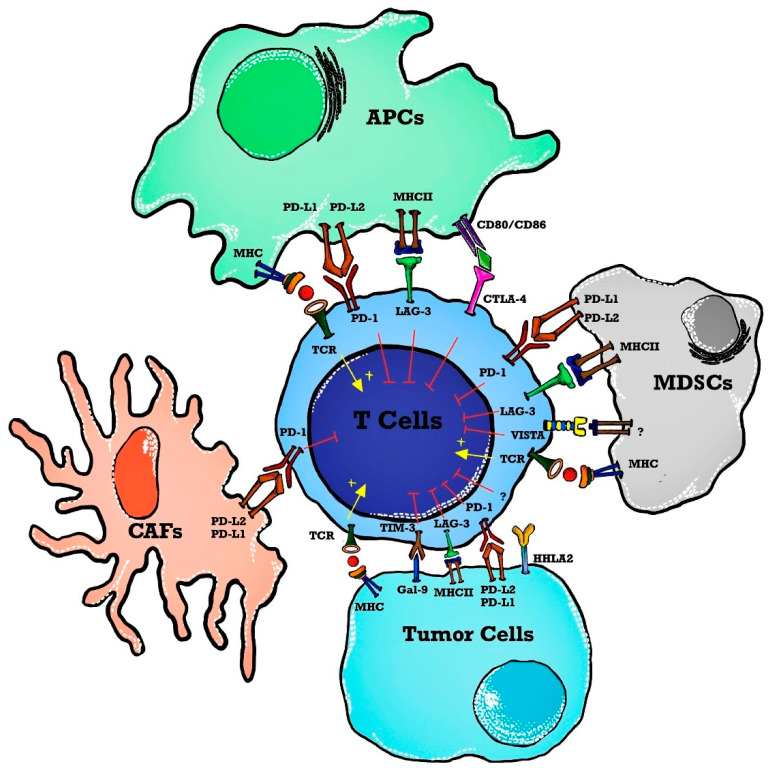
Immune checkpoints in a tumor microenvironment (TME). T cell activation is induced as APCs present tumor antigens to naïve T cells. The MHC and TCR signaling pathway provides a signal for T cell activation whereas immune checkpoints inhibit T cell activation in TME. Immune checkpoint markers are expressed on T cells, and ligands are present on APCs, tumor cells, and stromal cells such as CAFs and MDSCs. Abbreviations: APCs—antigen-presenting cells; MHC—major histocompatibility complex; TCR—T cell receptor; TME—tumor microenvironment; MDSCs—myeloid-derived suppressor cells; PD-1—programmed death 1; PD-L2—programmed cell death ligand-2; VISTA—V-domain immunoglobulin containing suppressor T cell activation; HHLA2—human endogenous retrovirus-H long terminal repeat-associated protein 2; TIM-3—T cell immunoglobulin and mucin domain 3; Gal-9—Galedctin-9; CAFs—cancer-associated fibroblasts; LAG-3—lymphocyte activated gene-3; CTLA-4—cytotoxic T-lymphocyte antigen-4.

**Figure 2 ijms-24-12502-f002:**
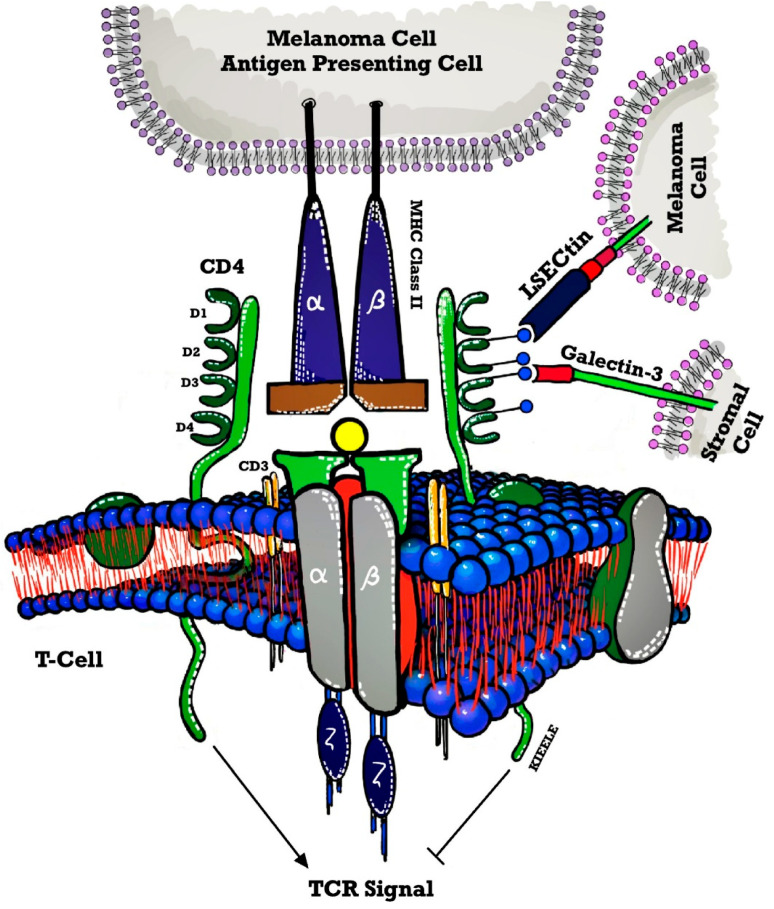
Structural similarities between LAG-3 and CD4. The LAG-3 gene is predicted to be highly structurally homologous to CD4 with four extracellular immunoglobulin superfamily (IgSF) namely D1-D4. LAG-3 binds to MHC class II with high affinity. LAG3 cytoplasmic domain appears to have three well-defined motifs namely a serine-based motif that could act as a PKC substrate, an “EP” motif made up of a series of glutamic acid-proline dipeptide repeats, and a relatively unique “KIEELE” motif, highlighted by an essential lysine residue. LAG-3 has two additional ligands namely LSECtin expressed on melanoma cells and Galectin-3 expressed on stromal cells and CD8+ T cells in TME. Abbreviations: TCR Toll-like receptor.

**Figure 3 ijms-24-12502-f003:**
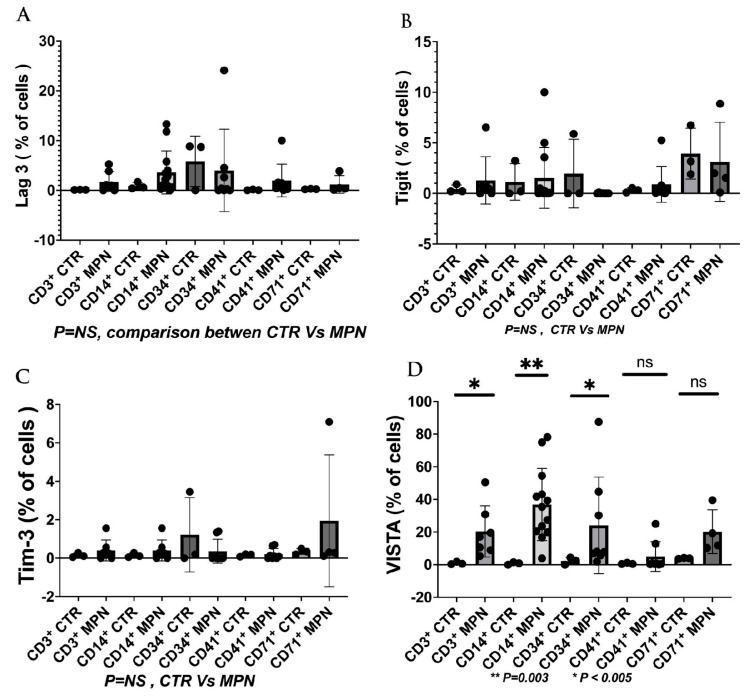
2nd generation of ICI expressions (% of cells) on the different cell populations were done on MPN patients and controls. There were 5 controls and 13 patients (**A**). The results showing that there was no difference on the LAG-3 (**B**). Also, there was no difference on the TIGIT between MPN and control (**C**). As well as no difference on the TIM-3 on the different cell population between MPN and controls (**D**). A significance of the VISTA expression (between MPN and controls) (mean + SE) on the CD3 (20.4 + 5.94 vs. 0.91 + 0.44, *p* < 0.05) CD14 (38.86 + 6.12 vs. 0.79 + 0.43, *p* = 0.003), CD 34 (2.30 + 1.26 vs. 2.30 + 1.28, *p* < 0.05), other ICI marker of LAG3 and TIM3 were of no significant difference. (Note: the significant values are highlighted with an asterisk).

**Figure 4 ijms-24-12502-f004:**
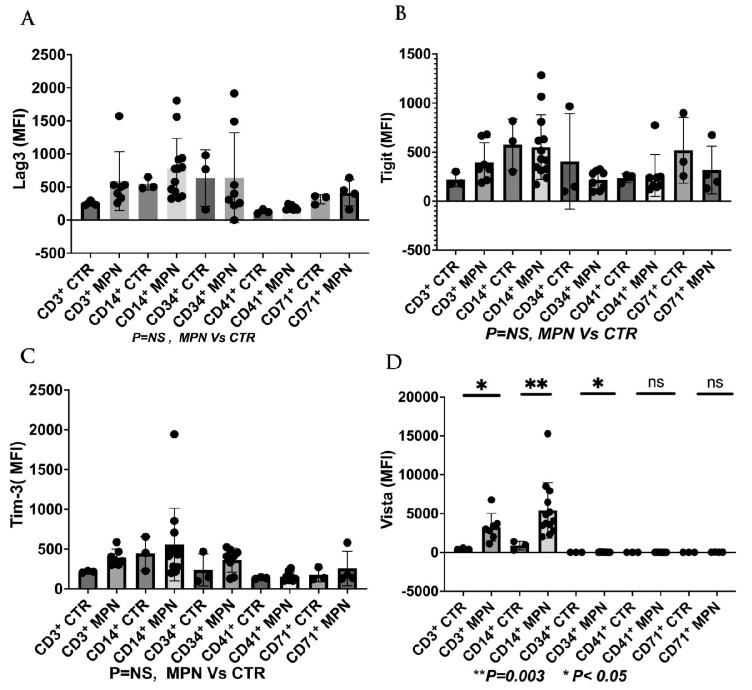
2nd generation of ICI expression (MFI) on different cell populations on MPN patients and controls (**A**). The results showing there was no differences of the LAG-3 (**B**). Showing no differences of the TIGIT on the different cell populations between MPN and controls (**C**). Showing no differences of the TIM-3 on the different cell population between MPN and controls (**D**). There was a significance of the VISTA expression (between MPN and controls) (mean + SE) on the CD3 (3257 + 673.4 vs. 457.0 + 59.0, *p* < 0.05) CD14 (5399 + 994.3 vs. 879.3 + 325.2, *p* = 0.003), CD 34 (2300 + 1262 vs. 24.0 + 10.4, *p* < 0.05).

**Figure 5 ijms-24-12502-f005:**
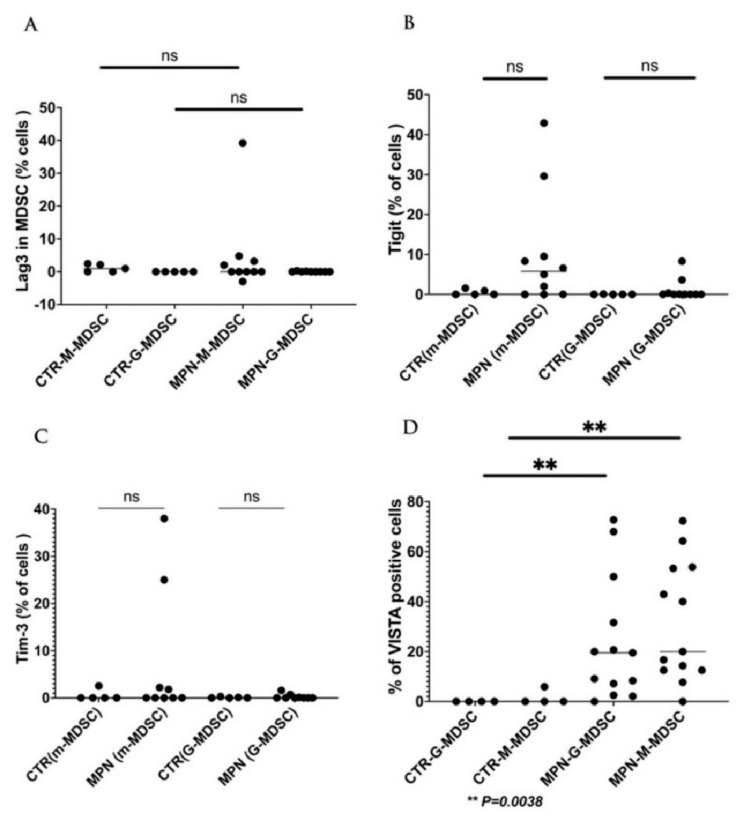
Expression of 2nd generation of ICI (% of positive cells) on the G-MDSC, and M-MDSC in patients with MPN and controls (**A**). The results showing no difference of the LAG-3 on G-MDSC or M-MDSC between MPN and controls (**B**). Showing no difference of the TIGIT on G-MDSC or M-MDSC between MPN and controls (**C**). Showing no difference of the TIM-3 on G-MDSC or M-MDSC between MPN and controls (**D**). Showing a significance of the VISTA expression (between MPN and controls) (mean + SE) on the G-MDSC (23.9 + 6.8 vs. 0.00 + 0.0, *p* < 0.003), and M-MDSC (31.5 + 6.6 vs. 1.47 + 1.47, *p* = 0.003) respectively.

**Figure 6 ijms-24-12502-f006:**
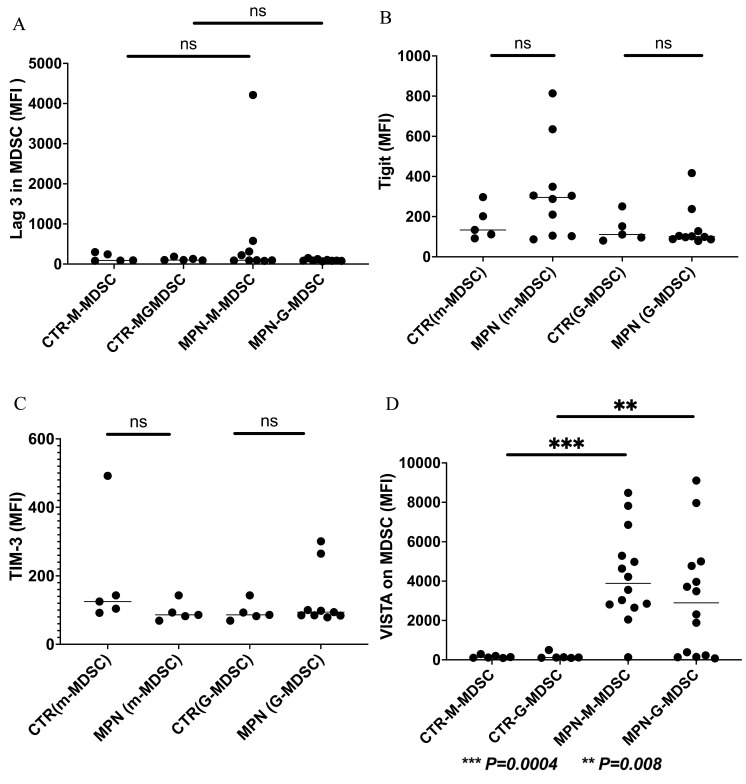
Expression of 2nd generation of ICI (MFI) on the G- MDSC, and M-MDSC in patients with MPN and controls (**A**). The results showing no difference of the LAG-3 on G-MDSC or M-MDSC between MPN and controls (**B**). Showing no difference of TIGIT on G-MDSC or M-MDSC between MPN and controls (**C**). Showing no difference of TIM-3 on G-MDSC or M-MDSC between MPN and controls (**D**). Showing a significance of the VISTA expression (between MPN and controls) (mean + SE) on the G-MDSC (3085 + 763.6 vs. 179.7 + 64.6, *p* = 0.008), and M-MDSC (4241 + 617.7 vs. 159.7 + 31.29, *p* = 0.0004) respectively.

**Table 1 ijms-24-12502-t001:** Summarizes all the clinical trials of LAG-3 therapy being used in hematological or related malignancies.

NCT Number	Sponsors/Collaorators	Title	Drug Name	Phase	Tumor/Disease	Status	Outcome Measures	ICI Type	Study Designs
NCT04566978	Memorial Sloan Kettering Cancer Center	89Zr-DFO-REGN3767 in PET Scans in People With Diffuse Large B Cell Lymphoma (DLBCL)	Drug: 89Zr-DFO-REGN3767|Diagnostic Test: PET/CT	Early Phase 1	Large B-cell Lymphoma|DLBCL	Recruiting	Biodistribution of 89Zr-DFO-REGN3767|Optimal 89Zr-DFO-REGN3767 mass dose for tumor targeting|Optimal time for imaging and tumor uptake post 89Zr-DFO-REGN3767 administration|Tumor lesion uptake of 89Zr-DFO-REGN3767 and correlate with LAG-3 expression by IHC	LAG-3	Allocation: Randomized|Intervention Model: Sequential Assignment|Masking: None (Open Label)|Primary Purpose: Diagnostic
NCT03310619	Celgene	A Safety and Efficacy Trial of JCAR017 Combinations in Subjects With Relapsed/Refractory B-cell Malignancies (PLATFORM)	Biological: JCAR017|Drug: Durvalumab|Drug: CC-122|Drug: Ibrutinib|Drug: CC-220|Drug: Relatlimab|Drug: Nivolumab|Drug: CC-99282	Phase 1|Phase 2	Lymphoma, Non-Hodgkin|Lymphoma, Large B-Cell, Diffuse|Lymphoma, Follicular	Completed	Dose-limiting toxicity (DLT) rates|Complete Response Rate|Adverse Events (AEs)|Progression-free survival (PFS)|Overall survival (OS)|Overall response rate (ORR)|Duration of response (DOR)|Event-free survival (EFS)|Pharmacokinetic (PK)-Cmax|Pharmacokinetic (PK)-Tmax|Pharmacokinetic (PK)-AUC|Health-related quality of life (HRQoL)|Quality of Life C30 questionnaire (EORTC-QLQ-C30)|European Quality of Life-5 Dimensions health state classifier to 5 Levels (EQ-5D-5L)	LAG-3	Allocation: Randomized|Intervention Model: Parallel Assignment|Masking: None (Open Label)|Primary Purpose: Treatment
NCT02061761	Bristol-Myers Squibb	A Phase 1/2a Dose Escalation and Cohort Expansion Study of the Safety, Tolerability, and Efficacy of Anti-LAG-3 Monoclonal Antibody (Relatlimab, BMS-986016) Administered Alone and in Combination With Anti-PD-1 Monoclonal Antibody (Nivolumab, BMS-936558) in Relapsed or Refractory B-Cell Malignancies	Biological: BMS-986016|Biological: BMS-936558	Phase 1|Phase 2	Hematologic Neoplasms	Completed	Safety measured by the rate of Adverse events (AEs), Serious Adverse events (SAEs), death and laboratory abnormalities [Time Frame: Up to approximately 2.3 years]	LAG-3	Allocation: Non-Randomized|Intervention Model: Single Group Assignment|Masking: None (Open Label)|Primary Purpose: Treatment
NCT03598608	Merck Sharp & Dohme LLC	A Phase 1/Phase 2 Clinical Study to Evaluate the Safety and Efficacy of a Combination of MK-4280 and Pembrolizumab (MK-3475) in Participants With Hematologic Malignancies	Biological: pembrolizumab|Biological: Favezelimab	Phase 1|Phase 2	Hodgkin Disease|Lymphoma, Non-Hodgkin|Lymphoma, B-Cell	Recruiting	Percentage of Participants Experiencing a Dose-limiting Toxicity (DLT); Percentage of Participants Experiencing an Adverse Event (AE); Percentage of Participants with Treatment Discontinuations Due to an AE	LAG-3	Allocation: Non-Randomized|Intervention Model: Parallel Assignment|Masking: None (Open Label)|Primary Purpose: Treatment
NCT04913922	Ludwig-Maximilians-University of Munich	An Open-Label Phase II Study of Relatlimab (BMS-986016) With Nivolumab (BMS-936558) in Combination With 5-Azacytidine for the Treatment of Patients With Refractory/Relapsed Acute Myeloid Leukemia and Newly Diagnosed Older Acute Myeloid Leukemia Patients	Drug: Azacitidine Injection|Drug: Nivolumab|Drug: Relatlimab	Phase 2	Acute Myeloid Leukemia	Recruiting	Maximum tolerated dose (MTD); Dose-limiting toxicities (DLTs); Objective response rate (ORR)	LAG-3	Allocation: N/A|Intervention Model: Single Group Assignment|Masking: None (Open Label)|Primary Purpose: Treatment
NCT03365791	Novartis Pharmaceuticals	Modular Phase 2 Study to Link Combination Immune-therapy to Patients With Advanced Solid and Hematologic Malignancies. Module 9: PDR001 Plus LAG525 for Patients With Advanced Solid and Hematologic Malignancies.	Biological: PDR001; Biological: LAG525	Phase 2	Small cell lung cancer, Gastric/esophageal adenocarcinoma, Castration resistant prostate adenocarcinoma (CRPC), Soft tissue sarcoma, Ovarian adenocarcinoma, Advanced well-differentiated neuroendocrine tumors, Diffuse large B cell lymphoma (DLBCL).	Completed	Clinical Benefit Rate (CBR) at 24 Weeks of PDR001+LAG525 by Tumor Type in Multiple Solid Tumors and Lymphoma	LAG-3	Allocation: N/A|Intervention Model: Single Group Assignment|Masking: None (Open Label)|Primary Purpose: Treatment

**Table 2 ijms-24-12502-t002:** Shows current clinical trials of anti-TIGIT antibodies in hematological malignancies.

NCT Number	Sponsors/Collaorators	Title	Drug Name	Phase	Tumor/Disease	Status	Outcome Measures	ICI Type	Study Designs
NCT05315713	Hoffmann-La Roche	A Phase Ib/II Open-Label, Multicenter Study Evaluating the Safety, Efficacy, and Pharmacokinetics of Mosunetuzumab in Combination With Tiragolumab With or Without Atezolizumab in Patients With Relapsed or Refractory B-Cell Non-Hodgkin Lymphoma	Mosunetuzumab SC; Tiragolumab; Atezolizumab; Tocilizumab	Phase 1 and 2	Relapsed or refractory (R/R) diffuse large B-cell lymphoma (DLBCL) or follicular lymphoma (FL)	Active, not recruiting	Percentage of Participants with Adverse Events; Best Objective Response Rate (ORR)	TIGIT	Allocation: Non-randomized|Intervention Model: Single Group Assignment|Masking: None (Open Label)|Primary Purpose: Treatment
NCT04045028	Genentech, Inc.	A Phase Ia/Ib Open-Label, Multicenter Study Evaluating the Safety and Pharmacokinetics of Tiragolumab as a Single Agent and in Combination With Atezolizumab and/or Daratumumab in Patients With Relapsed or Refractory Multiple Myeloma, and as a Single Agent and in Combination With Rituximab in Patients With Relapsed or Refractory B-Cell Non-Hodgkin Lymphoma	Tiragolumab; Daratumumab/rHuPH20; Rituximab; Atezolizumab	Phase 1	Relapsed or Refractory Multiple Myeloma, Relapsed or Refractory B-Cell Non-Hodgkin Lymphoma	Terminated (slow recruitment)	Percentage of Participants With Adverse Events; Objective Response Rate (ORR) for R/R MM; ORR for R/R NHL; Percentage of Participants With Anti-Drug Antibodies (ADAs) to Tiragolumab/Atezolizumab	TIGIT	Allocation: Non-randomized|Intervention Model: Parallel Assignment|Masking: None (Open Label)|Primary Purpose: Treatment
NCT05267054	BeiGene	A Phase 1b/2 Study Investigating the Safety, Tolerability, Pharmacokinetics, and Preliminary Antitumor Activity of Ociperlimab (BGB A1217) in Combination With Tislelizumab (BGB A317) or Rituximab in Patients With Relapsed or Refractory Diffuse Large B Cell Lymphoma	Ociperlimab; Tislelizumab; Rituximab	Phase 1 and 2	Relapsed Diffuse Large B-cell Lymphoma	Recruiting	Number of participants with adverse events (AEs); Recommended Phase 2 dose (RP2D) of ociperlimab when administered in combination with tislelizumab or rituximab; ORR; CRR; DOR; TTR; PFS; OS; Host immunogenecity	TIGIT	Allocation: Non-randomized|Intervention Model: Parallel Assignment|Masking: None (Open Label)|Primary Purpose: Treatment
NCT04150965	Multiple Myeloma Research Consortium	A Phase I/II Assessment of Combination Immuno-Oncology Drugs Elotuzumab, Anti-LAG-3 (BMS-986016) and Anti-TIGIT (BMS-986207)	Elotuzumab, pomalidomide, dexamethasone; Anti-LAG-3; Anti-LAG-3 + Pomalidimide + Dexamethasone; Anti-TIGIT; Anti-TIGIT + Pomalidimide + Dexamethasone	Phase 1 and 2	Relapsed Diffuse Large B-cell Lymphoma; Multiple Myeloma	Recruiting	Overall Response Rate; Frequency, type and grade of Adverse Events and Serious Adverse Events	TIGIT	Allocation: Randomized|Intervention Model: Sequential Assignment|Masking: None (Open Label)|Primary Purpose: Treatment
NCT05005442	Merck Sharp & Dohme LLC	A Phase 2, Open-label Study to Evaluate the Safety and Efficacy of MK-7684A (MK-7684 [Vibostolimab] With MK-3475 [Pembrolizumab] Coformulation) in Participants With Relapsed or Refractory Hematological Malignancies	Biological: Pembrolizumab/vibostolimab coformuation	Phase 2	Hematological Malignancies	Recruiting	Number of Participants with a Dose-Limiting Toxicity (DLT); Number of Participants Who Experienced an Adverse Event (AE); ORR, DOR, DCR	TIGIT	Allocation: NA|Intervention Model: Single group assignment|Masking: None (Open Label)|Primary Purpose: Treatment
NCT04354246	Compugen Ltd	A Phase 1 Study of The Safety and Tolerability of COM902 in Subjects With Advanced Malignancies	COM902 monotherapy; COM902 in combination with COM701 (both at the RDFE); Triplet combination of COM902 + COM701 + Pembrolizumab.	Phase 1	Advanced cancer; ovarian cancer; lung cancer; plasma cell neoplasm; MM; HNSCC; Microsatellite stabel colorectal carcinoma; MSS-CRC	Recruiting	The safety and tolerability of COM902 monotherapy; To identify the maximum tolerated dose (MTD) and/or recommended Phase 2 dose (RP2D); To characterize the pharmacokinetic (PK) profile of COM902 as monotherapy in subjects with advanced malignancies;	TIGIT	Allocation: Non-randomized|Intervention Model: Sequential Assignment|Masking: None (Open Label)|Primary Purpose: Treatment
NCT04254107	Seagen Inc.	A Phase 1 Study of SEA-TGT (SGN-TGT) in Subjects With Advanced Malignancies	SEA-TGT; sasanlimab; brentuximab vedotin	Phase 1	NSCLC; Gastric carcinoma; Gastroesophageal Junction Carcinoma; classic HL; DLBCL; Peripheral T-cell Lymphoma; Cutaneous Melanoma; Head and Neck Squamous Cell Carcinoma; Bladder carcinoma; ovarian cancer; triple negative breast cancer; cervical cancer	Recruiting	Number of participants with adverse events (AEs); Number of participants with laboratory abnormalities by grade; Number of participants with a dose-limiting toxicity (DLT) at each dose level	TIGIT	Allocation: Non-randomized|Intervention Model: Sequential Assignment|Masking: None (Open Label)|Primary Purpose: Treatment

## Data Availability

No data study was conducted for this research.

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
