# Peer review of "Role of Next Generation Immune Checkpoint Inhibitor (ICI) Therapy in Philadelphia Negative Classic Myeloproliferative Neoplasm (MPN): Review of the Literature"

_ijms, 2023, doi:10.3390/ijms241512502_

Round 1

Reviewer 1 Report

This is a well-written review article on a clinically-relevant topic. The information is well-organized through tables and figures. The inclusion of detailed information on clinical trials is helpful. I support the publication of this article. 

Author Response

Thank you for reviewing the article. We appreciate reviewer's time and consideration for the same. There are no changes/edit suggested by the reviewer hence no changes done in the article. 

Reviewer 2 Report

- The authors present a detailed review of LAG-3 , TIM-3 and VISTA and their role in hematological malignancies. However, the authors list additional immune checkpoints in TME from line 92 - 97. I suggest the authors to present known literature about the other listed immune check point inhibitors in relation to MPNs. 

- Majority of the review is focused on the association of mentioned genes in hematological malignancies. I suggest the authors to add commentary on the hypothesis relating the role of these genes in MPNs and specifically in philadelphia negative MPN cases.

- The authors show the expression of these genes through preliminary studies. I suggest the authors to add commentary or present data from published next-generation sequencing studies with regards to the genes presented . 

Author Response

Thank you for the review. Here are my answers:

1) "I suggest the authors to present known literature about the other listed immune check point inhibitors in relation to MPNs"

Reply: Although there are several next generation ICIT mentioned in lines 92-96 but our article is concentrating mainly on the use of LAG-3, TIM-3, and VISTA in classical MPN as our lab has preliminary data for the aforementioned ICIT.

2) "I suggest the authors to add commentary on the hypothesis relating the role of these genes in MPNs and specifically in philadelphia negative MPN case"

Reply: The review is about next generation ICIT in classical MPN and the emphasis has been on their therapeutic benefits and clinic implications. Details of genes was not part of the review.

3) "I suggest the authors to add commentary or present data from published next-generation sequencing studies with regards to the genes presented".

Reply: The article highlights the role of ICIT in hematological malignancies and the emphasis on clinical use. Sequencing studies of the genes can be added in future article while studying the genetic pleomorphism and their phenotypic expressions.

Reviewer 3 Report

While this review is dedicated to very interesting subject, authors should focus more on the main idea of the review. Most of the segments contain redundand information with lack of conscise presentation of the ICI+MPN topic. 

Major:

1) Figure 1. Illustrations are nice, however one may note that different types of biological molecules are designated with the same pictogram (i.e. TCR and CTLA4) and the same molecule with different pictograms (PD1) please correct and present data unifromly
2) LAG3 trials description is redundant. I dont see most of the information to be relevant.
3) The structure of segments is different, for exaple VISTA part contain table with only 3 trials, while numerous TIM3 inhibitors listed as a plain text. Consider revising uniformly

Minor:

- Please check the text for grammar and technical issues (for example replacement of hyphen with special symbols in some parts of the text ) 

- consider citing 10.1371/journal.pone.0275399

Author Response

Dear editor,

  1. I have taken your assessment and I have re-drawn the material to be uniform. Will resubmit the images that are now corrected.
  2. I will discuss with the rest of the authors to either revise or remove this section. Thank you very much for the input. 
  3. We will remove the table that has 3 trials and uniformly keep the same structure in plain text. 
  4. We will review again and make the correction to grammar as you mentioned.
  5. I will discuss this with the other authors to include the citation. 

Thank you very much for your review, we truly appreciate your feedback.